# Pancancer Analysis of the Oncogenic and Prognostic Role of *NOL7*: A Potential Target for Carcinogenesis and Survival

**DOI:** 10.3390/ijms23179611

**Published:** 2022-08-25

**Authors:** Qiaojun Liu, Renjian Xie, Yumei Li

**Affiliations:** 1School of Basic Medicine, Gannan Medical University, Ganzhou 341000, China; 2Key Laboratory of Biomaterials and Biofabrication in Tissue Engineering of Jiangxi Province, Ganzhou 341000, China; 3Key Laboratory of Prevention and Treatment of Cardiovascular and Cerebrovascular Diseases, Ministry of Education, Gannan Medical University, Ganzhou 341000, China; 4School of Medical Information Engineering, Gannan Medical University, Ganzhou 341000, China

**Keywords:** *NOL7*, pancancer, oncogenesis, prognosis, human tumor, enrichment analysis, big data

## Abstract

Despite growing evidence suggesting the critical function of *NOL7* in cancer initiation and development, a systematic pancancer analysis of *NOL7* is lacking. Herein, we present a comprehensive study of *NOL7* which aimed to explore its potential role and detailed mechanisms across 33 human tumors based on The Cancer Genome Atlas (TCGA) and Clinical Proteomic Tumor Analysis Consortium (CATPAC) databases. As a result, both gene and protein levels of *NOL7* were found to be increased in various tumor tissues, including breast invasive carcinoma (BRCA), colon adenocarcinoma (COAD), hepatocellular carcinoma (LIHC), lung adenocarcinoma (LUAD), and head and neck squamous cell carcinoma (HNSC) as compared with corresponding normal tissues. Meanwhile, dysregulated *NOL7* expression was found to be closely related to pathological stage and prognosis in several cancers, including LIHC, ovarian serous cystadenocarcinoma (OV), and bladder urothelial carcinoma (BLCA). The DNA methylation level of *NOL7* was found to be decreased in most cancers and to be negatively associated with *NOL7* expression. Furthermore, *NOL7* expression was determined to be significantly associated with levels of infiltrating cells and immune checkpoint genes, including *HMGB1*. Analysis of *NOL7*-related genes revealed that RNA metabolism pathways, including “ribosome biogenesis”, “spliceosome”, and “RNA transport”, were mainly involved in the functional mechanism of *NOL7* in human cancers. In summary, this pancancer study characterized the relationship between *NOL7* expression and clinicopathologic features in multiple cancer types and further showed its potential regulatory network in human cancers. It represents a systemic analysis for further functional and therapeutic studies of *NOL7* and highlights its predictive value with respect to the carcinogenesis and prognosis of various cancers, especially LIHC.

## 1. Introduction

Targeted therapy has become mainstream and one of the most effective strategies for cancer treatment [1]. However, the curability and prognosis of patients remain unsatisfactory due to an insufficiency of targeted drugs. The therapeutic effects of current targeted drugs are also limited by the emergence of drug resistance [2]. Therefore, it is necessary to explore novel and unique therapeutic targets for cancer treatment. Pancancer analysis represents a systemic approach to the characterization of genes of interest in multiple cancers. It is a promising method for investigating the relationship between genes or molecules and potential related theories of carcinogenesis as well as cancer progression. It can also provide a theoretical reference for the development of new tumor-targeting drugs and further expansion of the population who may benefit from targeted therapy [3]. 

The human *NOL7* gene is located at 6p23, a chromosomal region frequently lost in several malignancies, including cervical cancer [4]. It was first reported in 2006 by Lingen MW et al. that *NOL7* serves as a tumor suppressor gene in cervical cancer [5]. Both loss of heterozygosity and decreased mRNA/protein expression of *NOL7* were observed in cervical cancer, and it was found to be downregulated with disease progression. Meanwhile, *NOL7* can predict cancer recurrence and survival in cervical cancer patients. Researchers found that tumor-associated mutations and SNP variations harbored in the *NOL7* gene potentially contribute to the loss of *NOL7* in cancer cells [6,7]. In addition, C-myc and RXRα transcription factors were found to positively regulate *NOL7* expression [4]. MW Lingen et al. also revealed that transcription of *NOL7* was positively regulated by the *RB* tumor suppressor in cervical cancer and that it further functioned as an mRNA-binding protein to modulate TSP-1 expression by enhancing the post-transcriptional stability of its mRNA, thus inducing an anti-angiogenic and tumor-suppressive phenotype [8,9]. The *NOL7* gene encodes nucleolar protein 7, which has three alternatively spliced forms, each isoform differently localized within the cell. *NOL7*-SP1 (29 kD) is predominantly localized in the cell nucleolus, with some nucleoplasmic distribution; *NOL7*-SP2 (16 kD) is nucleoplasmic and has a moderate nucleolar distribution; while *NOL7*-SP3 (12 kD) is mostly distributed in the nucleoplasm, with some cytoplasmic distribution. A strong nuclear localization signal within the C-terminus of *NOL7*-SP1 contributes to its translocation into the cell nucleus. Nucleolus-localized *NOL7* protein is typically involved in maintaining internal nucleolar structure and cell growth, whereas nucleoplasm-localized *NOL7* protein exerts anti-angiogenic effects [10,11]. It is well worth noting that the chromosome region 6p23, where the *NOL7* gene resides, frequently undergoes heterozygous loss in cervical cancer, and this region is commonly associated with the amplification of melanoma [12]. Thus, our previous research determined that *NOL7* was upregulated in melanoma development and exhibited tumor-promoting activity [13]. The multifaceted function of *NOL7* suggested that its investigation might enable mechanistic insights into the potential role of *NOL7* in malignancy. However, a systematic assessment of *NOL7* in human cancer remains unavailable. 

Given the critical role of *NOL7* in cancer, we first combined TCGA project, GTEx, and CPTAC datasets to conduct a pancancer analysis of *NOL7* in various human cancers. This comprehensive study aimed to clarify the potential function and regulatory mechanisms of *NOL7* in the pathogenesis or clinical prognosis of multiple cancer types and further confirm its promising role as a diagnostic biomarker and therapeutic target for cancer treatment.

## 2. Results

### 2.1. Pancancer Analysis of NOL7 Expression

As shown in Appendix A, the *NOL7* gene is localized at the 6p23 chromosomal region (chromosome 6: 13,615,335–13,632,739 forward strand). We downloaded data on the subcellular localization and tissue localization of the *NOL7* protein from the HPA database and found that the *NOL7* protein had the highest expression in the kidneys, followed by the pancreas (Appendix A). Overall, *NOL7* showed low RNA specificity in tissue as well as in blood cells (Appendix A). 

Moreover, a gene expression landscape of *NOL7* across human cancers was constructed based on TCGA project data. Figure 1a shows significant differences in the transcriptional levels of *NOL7* between tumor tissues versus adjacent normal tissues in 16 types of cancer. Compared with normal tissue, increased *NOL7* expression was observed in BLCA, BRCA, cholangiocarcinoma (CHOL), COAD, esophageal carcinoma (ESCA), head and neck squamous cell carcinoma (HNSC), LIHC, LUAD, lung squamous cell carcinoma (LUSC), prostate adenocarcinoma (PRAD), and stomach adenocarcinoma (STAD) (all *p*-values < 0.001). Conversely, decreased *NOL7* expression was detected in kidney chromophobe (KICH), kidney renal clear cell carcinoma (KIRC), kidney renal papillary cell carcinoma (KIRP), and thyroid carcinoma (THCA) tissues (all *p*-values < 0.001) as compared with normal tissues. We also incorporated the GTEx datasets to supplement the number of normal tissues included as controls in the TCGA project. As shown in Figure 1b, *NOL7* levels were significantly higher in thymoma (THYM), CHOL, DLBC, LIHC, and pancreatic adenocarcinoma (PAAD) tissues as compared with corresponding normal tissues, while they were markedly lower in acute myeloid leukemia (LAML) tissues as compared with normal tissues (all *p*-values < 0.05).

In addition, we utilized CPTAC datasets to detect the differences in *NOL7* protein levels between tumor tissues and corresponding normal tissues. The protein levels of *NOL7* were found to be notably increased in breast cancer, ovarian cancer, colon cancer, hepatocellular carcinoma, KIRC, adrenocortical carcinoma (RCC), LUAD, glioblastoma multiforme (GBM), and HNSC tissues (all *p*-values < 0.001) but decreased in PAAD tissues (*p* < 0.05) as compared with relevant healthy tissues (Figure 1c). The immunohistochemistry staining images of *NOL7* proteins in various tumors and corresponding normal tissues are shown in Appendix A.

### 2.2. Pancancer Analysis of the Relationship between NOL7 Expression and Clinicopathology

To investigate the association between NOL7 and clinicopathological features, we analyzed the expression patterns of NOL7 in normal tissues and stages Ⅰ, Ⅱ, Ⅲ, and Ⅳ tumor tissues in multiple cancers using the UALCAN tool. As shown in Figure 2, NOL7 expression was confirmed to be significantly upregulated during disease progression from normal tissue to early malignancy and further to the terminal stage in cancers, including BLCA, BRCA, CHOL, COAD, ESCA, HNSC, LIHC, LUAD, LUSC, READ, and STAD. It is worth noting that for KICH, KIRC, KIRP, and THCA, NOL7 expression was markedly downregulated during disease progression. However, NOL7 expression showed no significant change during disease progression in several cancers, including cervical squamous cell carcinoma (CESC), PAAD, skin cutaneous melanoma (SKCM), and uterine corpus endometrial carcinoma (UCEC). Furthermore, NOL7 protein expression was found to be significantly associated with disease progression in multiple cancers, including breast cancer, ovarian cancer, colon cancer, KIRC, HNSC, and LUAD (Appendix A).

### 2.3. Survival Analysis of NOL7 in the Pancancer Analysis

To further estimate the correlation between NOL7 expression and prognosis in human patients with cancers, the Kaplan–Meier plotter and GEPIA2 were utilized, respectively. According to the results from the Kaplan–Meier plotter, Figure 3a suggested that enhanced expression of NOL7 was associated with poor overall survival (OS) in ESCA (*p* < 0.05), KIRP (*p* < 0.001), LIHC (*p* < 0.01), sarcoma (SARC) (*p* < 0.001), and UCEC (*p* < 0.05). Among all tumor types, KIRP showed the greatest significance. In contrast, reduced NOL7 expression was related to favorable OS in bladder carcinoma (*p* < 0.05), KIRC (*p* < 0.01), ovarian cancer (*p* < 0.001), and READ (*p* < 0.001). Figure 3b illustrates that highly expressed NOL7 was negatively correlated with relapse-free survival (RFS) in KIRP, LIHC, and UCEC and positively correlated with RFS in bladder carcinoma (*p* < 0.001) and ovarian cancer (*p* < 0.01). In addition, increased NOL7 expression corresponded with poor distant metastasis-free survival (DMFS) (*p* < 0.05) and RFS (*p* < 0.001) in breast cancer but was related to better OS (*p* < 0.001), first progression (FP) (*p* < 0.001), and post-progression survival (PPS) (*p* < 0.001) in gastric cancer. Additionally, increased NOL7 expression was remarkably related to poor PFS (*p* < 0.001) and better PPS (*p* < 0.01) in ovarian cancer and to better FP (*p* < 0.001) in lung cancer (Appendix A). Moreover, subgroup analyses of breast cancer, lung cancer, gastric cancer, ovarian cancer, and liver cancer were conducted to investigate the association between NOL7 expression and the prognoses of patients in the different groups (Appendix A).

According to the results from GEPIA2, we observed that dysregulated NOL7 expression significantly affected OS in seven types of cancer. For KIRC and OV, a favorable prognosis was presented when NOL7 expression was increased (all *p*-values < 0.001); the indications were opposite for other cancers, including ACC, ESCA, KICH, SARC, and LIHC (all *p*-values < 0.05; Appendix A). Appendix A indicates that highly expressed NOL7 predicted poor DFS for ACC, CHOL, HNSC, LIHC, KICH, SKCM, and uterine carcinosarcoma (UCS) (all *p*-values < 0.05). However, augmented NOL7 expression predicted a good prognosis in terms of OS in both KIRC and OV (all *p*-values < 0.05). Taken together, these results demonstrated that NOL7 expression could predict patient survival in several cancer types, especially ACC, KICH, KIRC, LIHC, LUAD, and OV. However, the specific relationship between NOL7 expression level and prognosis in cancer patients depends on the tumor type.

### 2.4. Pancancer Analysis of DNA Methylation Levels and Genetic Alteration of NOL7

Abnormal DNA methylation was reported to lead to an increased risk of cancer [14]. Based on the TCGA project, we used UALCAN to explore the potential relationship between NOL7 DNA methylation level and tumorigenesis. A significant decrease in the methylation level of NOL7 was observed in HNSC, BLCA, ESCA, LUAD, LUSC, PRAD, pheochromocytoma and paraganglioma (PCPG), testicular germ cell tumors (TGCTs), THCA, UCEC (all *p*-values < 0.001), KIRP, and ESCA (all *p*-values < 0.01) as compared with normal tissues (Figure 4a). However, no differences were observed between BRCA, CHOL, COAD, CESC, GBM, KIRC, PAAD, READ, STAD, SARC, and THYM tissues and matching normal tissues (Appendix A).

We also determined the genetic variation in NOL7 using the cBioPortal database. With this approach, we found that, among the assessed cancers, OV bears the highest genetic alteration frequency (>6%) for NOL7. Amplification type variation accounted for a significant portion of the alterations in OV, DLBC, CHOL, uveal melanoma (UVM), UCS, and HNSC (Figure 4b). The investigation of the types and sites of NOL7 genetic alteration verified the Q239Tfs*7 frameshift insert mutation in three cases of uterine endometrioid carcinoma and one case of uterine endometrioid carcinoma. The above results indicate that this is a hotspot mutation site in cancer (Figure 4b). We further tested the potential relationship between genetic alterations in NOL7 and the prognosis of patients with different types of cancer. The results revealed that genetic alterations in NOL7 were related to poor DFS, but there were no significant differences in the prognoses of patients with differences in NOL7 alterations in terms of OS, disease-specific survival (DSS), or PFS (Figure 4c).

### 2.5. Pancancer Analysis of NOL7 Expression and Immune Cell Infiltration

Tumor-infiltrating immune cells play a key role in the progression and invasion of cancer [15]. Given that NOL7 is overexpressed in peripheral blood mononuclear cells (fold change compared to the average of all tissues), we performed a pancancer analysis to study the correlation between NOL7 expression and immune infiltration level. As shown in Figure 5, NOL7 expression was significantly correlated with the abundance of infiltrating immune cells: CD4+ T cells in 14 types of cancer, CD8+ T cells in 16 types of cancer, B cells in 11 types of cancer, neutrophils in 16 types of cancer, macrophages in 14 types of cancer, and DCs in 18 types of cancer. Additionally, NOL7 expression was positively and significantly correlated with these immune cells in KIRC, PRAD, KICH, LIHC, OV, and PAAD. CD8+ T cells and neutrophils were most strongly positively associated with NOL7 expression in these different cancers. Moreover, the results of the immune checkpoint analysis showed that NOL7 expression in most tumor tissues was positively correlated with C10orf54, CD276, IL12A, and especially HMGB1. The above results indicate that NOL7 may be a potential target for immune therapy (Appendix A).

We further explored the correlations between NOL7 expression and TMB and MSI. A positive correlation between NOL7 expression and MIS was observed in GBM, brain lower-grade glioma (LGG), stomach and esophageal carcinoma (STES), SARC, STAD, UCEC, HNSC, KIRC, LUSC, and UCS. In addition, NOL7 expression was positively correlated with TMB in LUAD, LAML, BRCA, ESCA, STES, SARC, STAD, LUSC, THYM, LIHC, PAAD, SKCM, UCS, BLCA, and ACC (Appendix A).

### 2.6. Gene Enrichment Analysis of NOL7 in the Pancancer Analysis

NOL7-interacting molecules and NOL7-correlated genes were selected to conduct GO and KEGG pathway enrichment analyses. We obtained the top 100 genes related to NOL7 expression in 33 human cancers based on GEPIA2. The top five related genes were MCUR1 (R = 0.6), EEF1E1 (R = 0.61), SSB (R = 0.57), DHX9 (R = 0.53), and WRNIP1 (R = 0.6) (all *p*-values < 0.01) (Figure 6a, above). The correlation heatmap data showed that the expression of NOL7 was positively correlated with the expression of these genes in the majority of detailed cancer types (Figure 6a, below). In addition, we used the STRING tool to collect the top 50 NOL7-interacting proteins, which identifications were supported by experimental evidence. The PPI network for these 50 proteins is shown in Figure 6b. Moreover, four common members were obtained by intersection analysis of NOL7-interacting and -correlated genes: NIFK, MPHOSPH10, NOL10, and WDR12 (Figure 6c). 

According to the results of the KEGG pathway analysis, “ribosome biogenesis”, “spliceosome”, and “RNA transport” were mainly involved in the effects of *NOL7* on tumor pathogenesis (Figure 6d). The GO enrichment analysis suggested that most of the NOL7-related genes were closely linked to biological processes, such as the RNA metabolic process, RNA processing, and ncRNA metabolic processes (Figure 6e). These results highlighted the central role of *NOL7* in intracellular molecule synthesis and cell proliferation.

## 3. Discussion

Existing studies have suggested the tumor-suppressive activity of *NOL7* in cervical cancer [5]. Our previous research first proposed an oncogenic role for *NOL7* in melanoma tumorigenesis and metastasis [13]. Based on the expected function of *NOL7* in human cancer, we conducted a pancancer analysis to characterize the expression landscape of NOL7 in various cancer types and further explore its prognostic value, genetic alterations, correlation with the cancer immune microenvironment, and regulatory network, aiming to better understand the potential features of *NOL7* in human cancers. 

Our study identified the aberrant expression (typically overexpression) of the *NOL7* gene in more than half of all cancer types (33 assessed in total) compared with normal tissues. Integration of these data with CPTAC datasets revealed that both gene and protein expression levels of *NOL7* were significantly enhanced in breast cancer, colon cancer, hepatocellular carcinoma, LUAD, and HNSC. Consistently, both gene and protein expression of *NOL7* were upregulated, along with disease progression in various cancers, including colon cancer, LUAD, and HNSC (Figure 1 and Figure 2 and Appendix A). These results suggested that *NOL7* may be pro-carcinogenic and function as a promising diagnostic biomarker for these cancers. It is notable that a potential contradictory of *NOL7* expression was observed in several cancers. For instance, the gene expression level of *NOL7* was found to be upregulated in PAAD, whereas its protein expression level was downregulated. The gene expression level of *NOL7* was decreased, along with disease progression in KIRC, but its protein expression level was increased (Figure 1 and Figure 2 and Appendix A). It is well established that genes do not act in isolation: the expression of gene-coded proteins is affected by many factors, including other genes, the environment, and epigenetics [16]. Crucially, mRNA transcription does not necessarily translate into protein expression, and it is not uncommon to observe a discrepancy between mRNA and protein expression [17]. Inconsistent expression at the gene level and at the protein level for *NOL7* in the same cancer type implies that the mechanisms of *NOL7* expression regulation are well worth exploring. The specific function of *NOL7* in different cancers depends on the particular cancer type, and gain- and loss-of-function approaches will hopefully be able to verify its exact functions. Furthermore, together with the results of the survival analysis performed with both the Kaplan–Meier plotter and GEPIA2, these results indicated the prognostic value of *NOL7* in various cancer types. Among them, highly expressed *NOL7* predicted unfavorable OS, RFS, and DFS in LIHC, whereas it predicted favorable OS, RFS, and DFS in OV. These results illustrated the meaningful prognostic value of *NOL7* with respect to survival in LIHC and OV (Figure 3 and Appendix A). Based on these observations, the critical function of *NOL7* in LIHC can be recognized, with the implication of oncogenic activities of *NOL7* in the carcinogenesis and cancer development of LIHC and the relevance of *NOL7* to patient survival. 

Previous researchers have confirmed that consistent loss of *NOL7* through loss of heterozygosity and decreased mRNA and protein expression has been observed in cervical cancer [7]. Additionally, the experimental study performed by Lingen MW et al. also confirmed the suppressive function of *NOL7* in cervical cancer [7]. However, the chromosome region 6p21–23, where *NOL7* resides, is frequently amplified in several cancers, including melanoma, and therefore may exert disease-promoting effects in these cancers [12]. In this study, significantly increased *NOL7* expression was found in metastatic sites as compared with primary sites in SKCM, suggesting the promotion of melanoma development. These findings were consistent with our previous conclusion [13], whereas gain in chromosome 6p21–23 is not frequently reported in hepatocellular carcinoma; thus, whether *NOL7* promotes carcinogenesis in hepatocellular carcinoma awaits verification [18]. 

Meanwhile, it was pointed out that nucleolus-localized *NOL7* protein is required to maintain internal nucleolar structure and cell proliferation [11]. This implies that the *NOL7*-encoding protein is an essential molecule for cytogenetic events and that loss of *NOL7* leads to the fatal destruction of cancer cells with uncontrollable proliferation properties. In addition, it has been reported that *NOL7* functions as an mRNA-binding protein [9,10]. GO enrichment analysis results also supported the previous findings and reflected a clearer and more exhaustive role for *NOL7* with respect to RNA, for instance, in RNA metabolic process and RNA processing. More importantly, KEGG pathway analysis results revealed that *NOL7*-related genes were mainly involved in ribosome biogenesis. Four genes that closely interacted with *NOL7* (NIFK, MPHOSPH10, NOL10, and WDR12) which were identified in our study were implicated in ribosome biogenesis (Figure 6). Ribosomes are responsible for protein synthesis in living cells. The essential roles of increased ribosome biogenesis and protein synthesis in sustaining tumor growth and progression are well established [19], and although a correlation between *NOL7* and ribosome biogenesis has not been reported, it is indicated that *NOL7* expression is strongly correlated with ribosome biogenesis. Furthermore, this relationship could be connected to *NOL7*’s tumor-promoting function. This finding suggested a novel regulatory mechanism for *NOL7* in cancer that deserves further investigation. Our study congruously supported these feasible molecular mechanisms. 

DNA methylation is a major epigenetic event that regulates gene expression without modifying DNA sequences [20]. The suppression or inactivation of tumor suppressor genes caused by the hypermethylation of DNA promoter regions is commonly reported in cancer cells [21]. Here, we have shown that the DNA methylation level of *NOL7* was downregulated in most types of cancer, which is consistent with the upregulation of *NOL7* (Figure 4 and Appendix A). Furthermore, the tumor immune microenvironment (TME) is a complex structure consisting of immune cells, endothelial cells, fibroblasts, and other substances. Interaction between cancer cells and various components of the TME favors the immune escape of tumors and eventually leads to the activated proliferation and invasion of cancer cells, which processes are related to tumor recurrence and the survival of patients [22,23]. The present study revealed that *NOL7* expression was significantly correlated with the infiltration levels of several types of immune cells, including B cells, CD8+ T cells, CD4+ T cells, macrophages, neutrophils, and DCs, in many cancers (Figure 6). This study also analyzed the relationship between *NOL7* expression and the expression of 60 common immune checkpoint genes, and notably found that *NOL7* expression was positively correlated with HMGB1 in most cancer types (Appendix A). HMGB1 encodes high mobility group box 1, which is a multifunctional molecule that participates in a variety of cellular biological properties, including DNA damage repair and maintenance of genomic stability, and is further involved in the regulation of the inflammatory response and the TME [24]. Our study suggests that *NOL7* might interact with HMGB1, thus functioning as a key mediator related to prognosis and the status of tumor immunity in cancers.

## 4. Materials and Methods

### 4.1. Gene Expression Analysis

The expression data for *NOL7* in different normal cells and tissues were downloaded from the HPA database directly. The sample source and ethical statement were provided at https://www.proteinatlas.org/ (accessed on 19 June 2022) [25]. Low specificity was defined by normalized expression (NX) ≥ 1 in at least one tissue/region/cell type but not elevated in any tissue/region/cell type. The expression differences between normal and tumor tissues in the TCGA database were assessed using TIMER2.0 (http://timer.cistrome.org/, accessed on 19 June 2022), based on the TCGA project [26,27]. 

Since several tumor tissues assessed with TIMER2.0 did not include adjacent normal tissues, we combined the data from the TCGA database with GTEx data for supplementary analyses of the expression differences between normal and tumor tissues through GEPIA2 (http://gepia2.cancer-pku.cn/, accessed on 19 June 2022) [28]. We selected the available tumor datasets, including LAML, THYM, CHOL, DLBC, LIHC, and PAAD, based on the significant differences in expression between tumor tissues and normal tissues. In addition, through the visualization of the CPTAC dataset via the UALCAN tool (http://ualcan.path.uab.edu/analysis-prot.html, accessed on 21 June 2022), we obtained the differences in total protein expression levels of *NOL7* between adjacent normal tissues and tumor tissues [29]. Furthermore, to explore the relationship between the expression of *NOL7* and tumor staging, we conducted a survey on UALCAN.

### 4.2. Survival Prognosis Analysis

The OS, RFS, FP, PFS, PPS, DMFS, and DSS of all tumor patients in the GEO cohorts were analyzed using the Kaplan–Meier plotter tool (http://kmplot.com/analysis/, (accessed on 24 June 2022) [30]. By setting the parameter to “auto select best cutoff”, patients were divided into two groups. Kaplan–Meier survival curves were obtained, and the log-rank *p*-values and HRs were calculated. Moreover, we analyzed the OS and DFS of all tumors using the “survival analysis” module of GEPIA2 (http://gepia.cancer-pku.cn/index.html, accessed on 19 June 2022) by setting the group cutoff to “median”. Patients were divided into a high group and a low group to generate Kaplan–Meier survival curves and survival maps and to calculate the log-rank *p*-values and HRs. 

### 4.3. DNA Methylation Analysis and Genetic Alteration Analysis

The UALCAN tool was used to explore the DNA methylation levels for *NOL7* between the various cancer tissues as compared with normal tissues. In addition, we chose the “TCGA Pan-Cancer Atlas Studies” module in the cBioPortal tool (https://www.cbioportal.org/, accessed on 1 July 2022) to analyze the genetic variation in *NOL7* [31]. Alteration frequency, mutation site, and survival data were downloaded by choosing “cancer types summary”, “mutations”, and “comparison/survival” in the query module.

### 4.4. Immune Infiltration Analysis

We used the Sangerbox tool (http://sangerbox.com/, accessed on 17 june 2022) to explore the association between *NOL7* expression and immune infiltration across 33 types of tumors by choosing the “immune cell analysis (TIMER)” module. In addition, the association between *NOL7* expression and many immune-checkpoint-related proteins was assessed by choosing the “immune checkpoint gene analysis” module in the same tool. Furthermore, we examined the relationship between *NOL7* expression and TMB or MSI in different tumors in the TCGA cohort [32]. Spearman’s correlation test was used to calculate *p*-values and partial correlation values. 

### 4.5. Gene Enrichment Analysis

To explore the regulatory network of *NOL7* in cancers, the STRING tool (https://cn.string-db.org/, accessed on 21 June 2022) was used to download the top 50 *NOL7*-interacting molecules. We visualized the PPI network using Cytoscape [33]. In addition, we obtained the top 100 *NOL7*-correlated genes through GEPIA2. Pearson’ s correlation test was performed on the top five *NOL7*-correlated genes in the “correlation analysis” module of GEPIA2 to obtain *p*-values, dot plots, a heatmap, and correlation coefficients. Moreover, we generated a Venn diagram, combining the two parts using the Draw Venn Diagram tool (http://bioinformatics.psb.ugent.be/webtools/Venn/, accessed on 21 June 2022) [34]. We combined these two sets of data and performed a GO enrichment analysis and a KEGG pathway analysis using the R package “cluster profile” in R software. A *p*-value < 0.05 and an FDR < 0.25 were considered statistically significant.

## 5. Conclusions

Overall, our first pancancer study enabled a relatively comprehensive understanding of the potential role of *NOL7* in various human cancers. We propose that *NOL7* may serve as a new diagnostic biomarker and therapeutic target in different tumor types, including breast cancer, colon cancer, lung cancer, and liver cancer. In addition, our work expands the understanding of the impact of *NOL7* on cancer immunologic therapy, as we revealed strong correlations between *NOL7* and immune cells and immune checkpoints. Future experimental and prospective studies of *NOL7* in different cancer types may provide in-depth insights into regulatory mechanisms and further support the development of therapeutic strategies targeting *NOL7*.

## Figures and Tables

**Figure 1 ijms-23-09611-f001:**
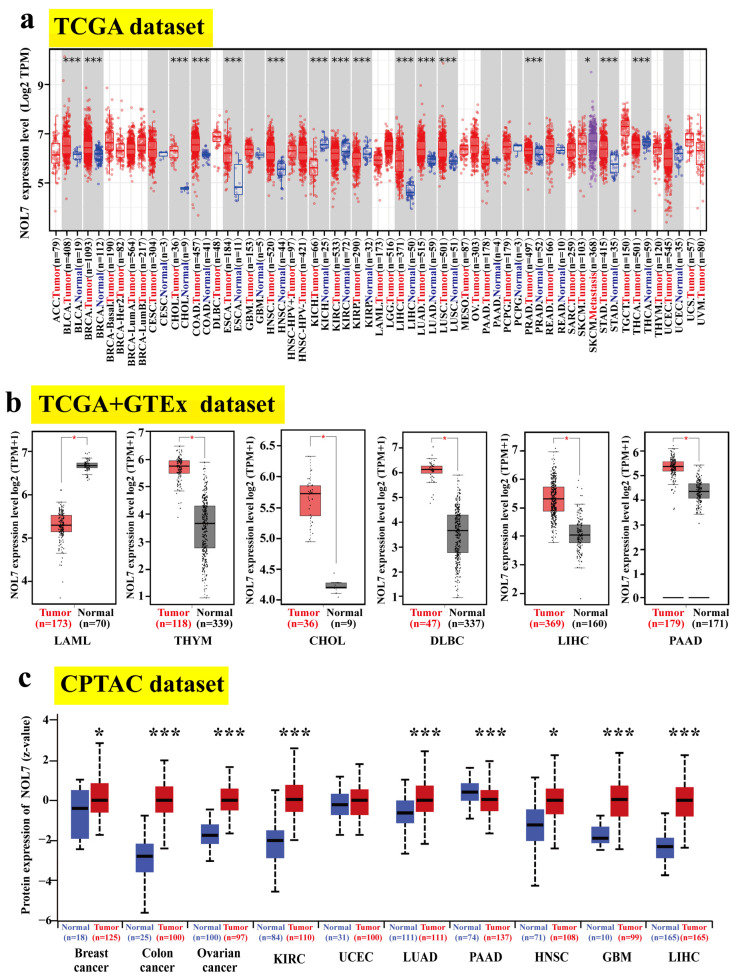
Expression status of *NOL7* in the pancancer analysis. (**a**) The expression levels of *NOL7* in tumor samples and paired normal tissues in different cancers in the TCGA cohort. (**b**) Combined with data for normal tissues in the GTEx database, a box plot representation of the comparison of *NOL7* expression levels in LAML, THYM, CHOL, DLBC, LIHC, and PAAD in the TCGA project database. (**c**) Comparison of *NOL7* protein expression levels in primary tissues and normal tissues of breast cancer, ovarian cancer, colon cancer, hepatocellular carcinoma, RCC, UCEC, LUAD, GBM, HNSC, and PAAD based on the CPTAC dataset. * *p* < 0.05; *** *p* < 0.001.

**Figure 2 ijms-23-09611-f002:**
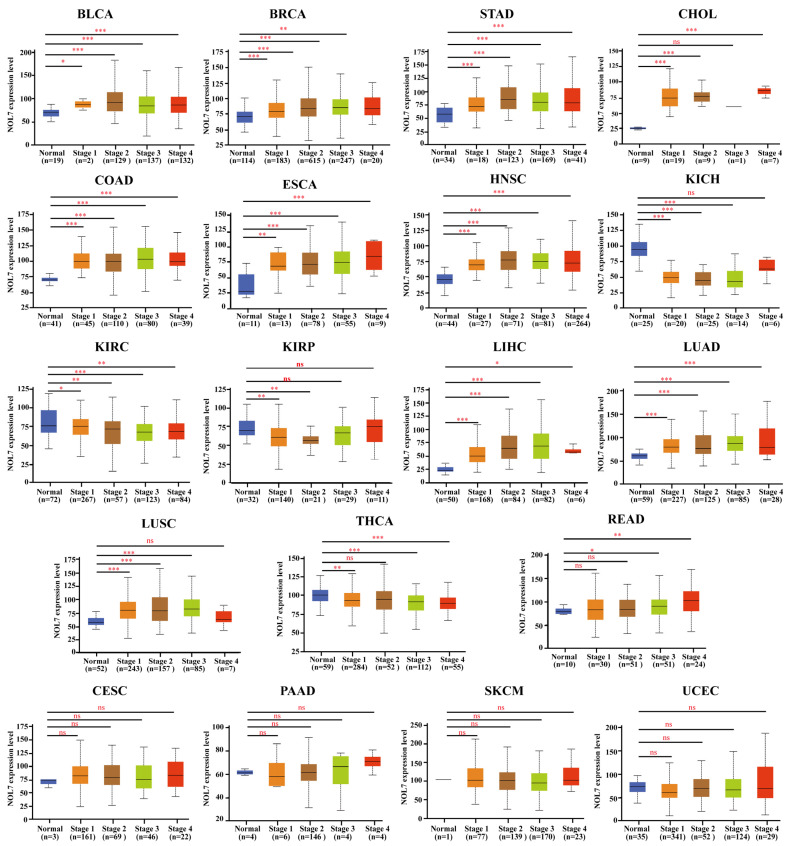
Correlations between *NOL7* gene expression and the main pathological stages in normal and stages I, II, III, and IV BLCA, BRCA, CESC, CHOL, COAD, ESCA, HNSC, KICH, KIRC, KIRP, LIHC, LUAD, LUSC, PAAD, READ, SKCM, STAD, THCA, and UCEC tissues, as examined using the TCGA dataset. * *p* < 0.05; ** *p* < 0.01; *** *p* < 0.001; ns No significance.

**Figure 3 ijms-23-09611-f003:**
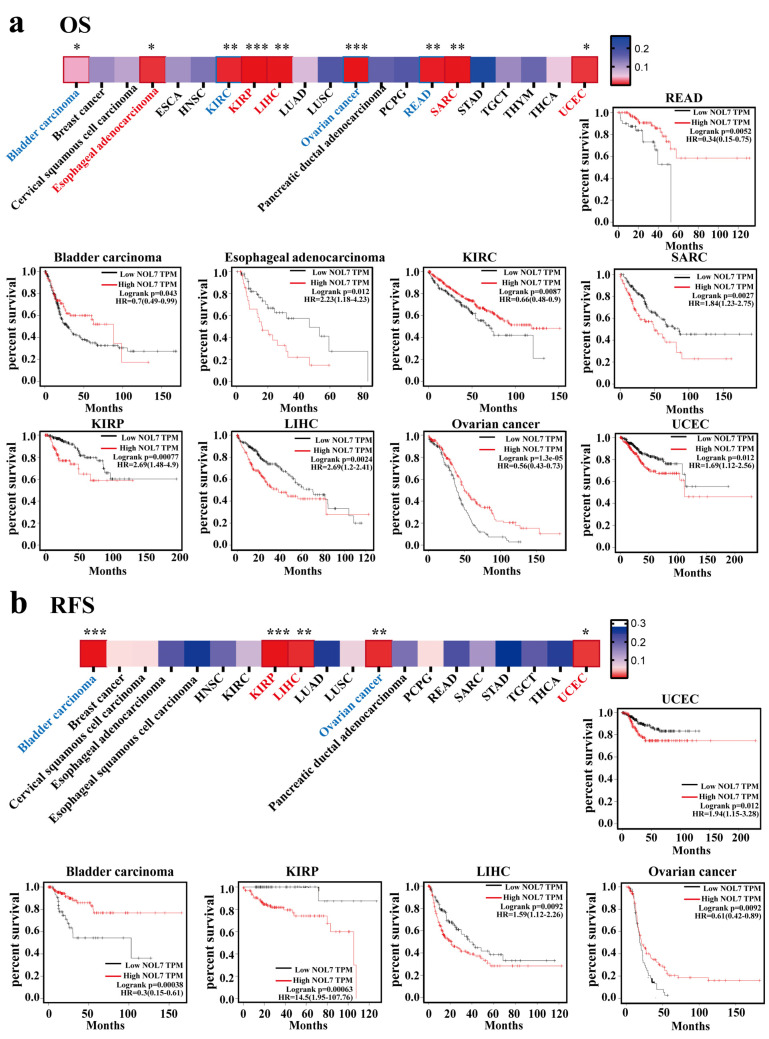
Survival prognoses of groups with high and low *NOL7* expression levels, according to the Kaplan–Meier plotter. (**a**) Correlation of OS with *NOL7* expression in bladder carcinoma, esophageal adenocarcinoma, KIRC, SARC, KIRP, LIHC, OV, UCEC, and READ. (**b**) Correlation of RFS with *NOL7* expression in bladder carcinoma, KIRP, LIHC, OV, and UCEC. * *p* < 0.05; ** *p* < 0.01; *** *p* < 0.001.

**Figure 4 ijms-23-09611-f004:**
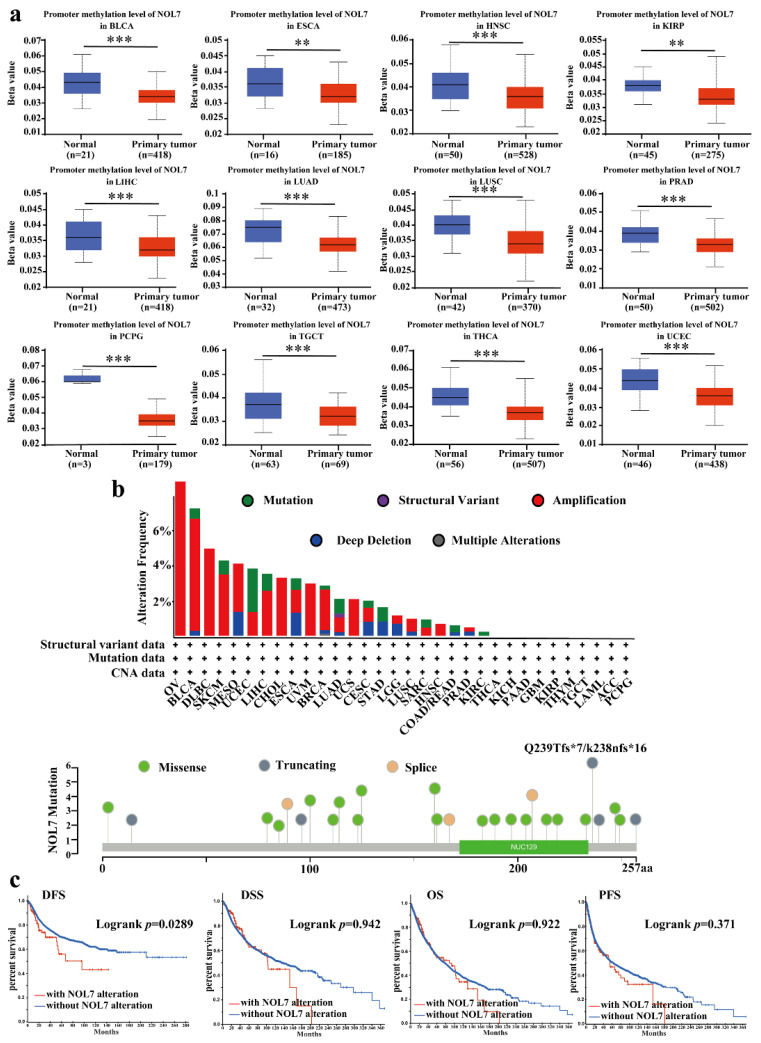
DNA methylation and mutation features of NOL7 in multiple tumors. (**a**) Promoter methylation levels of *NOL7* in BLCA, ESCA, HNSC, KIRP, ESCA, LUAD, LUSC, PRAD, PCPG, TGCT, THCA, and UCEC from the UALCAN database. (**b**) Alteration frequencies for different types of mutations of *NOL7* in the cBioportal database. The mutation types, sites, and case numbers of *NOL7* genetic alterations are shown below. (**c**) The effects of *NOL7* mutation status on OS, DFS, DSS, and PFS in cancer patients, according to data in the cBioPortal database. ** *p* < 0.01; *** *p* < 0.001.

**Figure 5 ijms-23-09611-f005:**
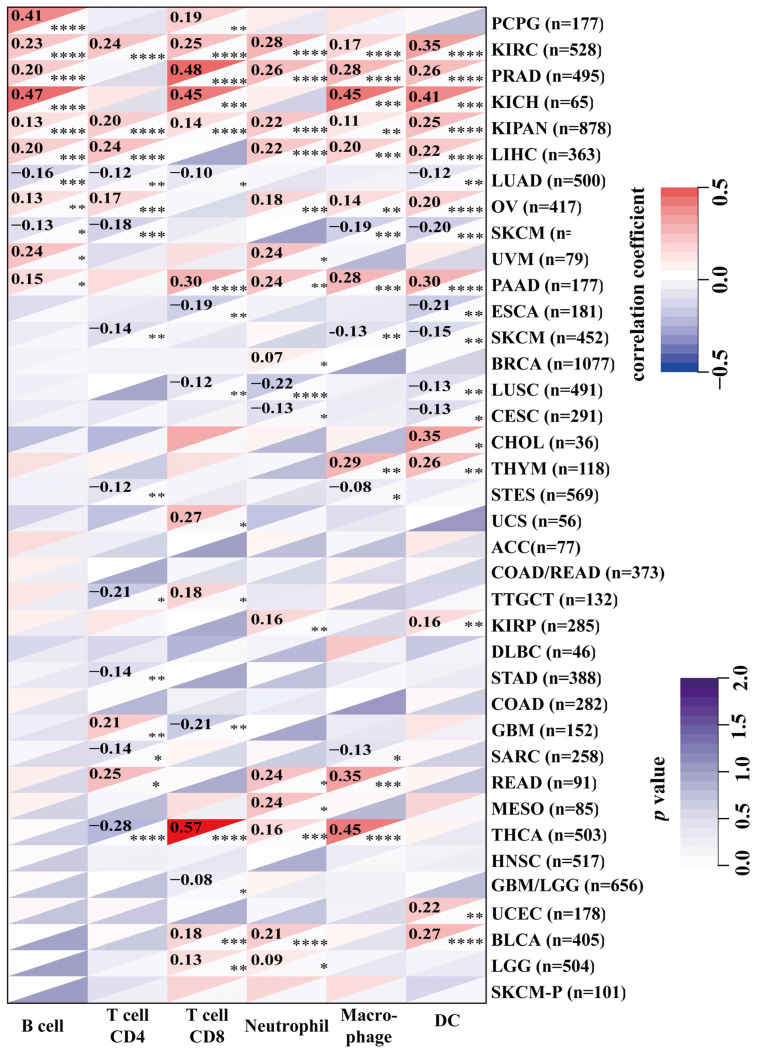
Association between *NOL7* expression and immune infiltration levels in various immune cells in the TIMER2.0 database, as assessed using the Sangerbox tool. * *p* < 0.05; ** *p* < 0.01; *** *p* < 0.001; **** *p*<0.0001.

**Figure 6 ijms-23-09611-f006:**
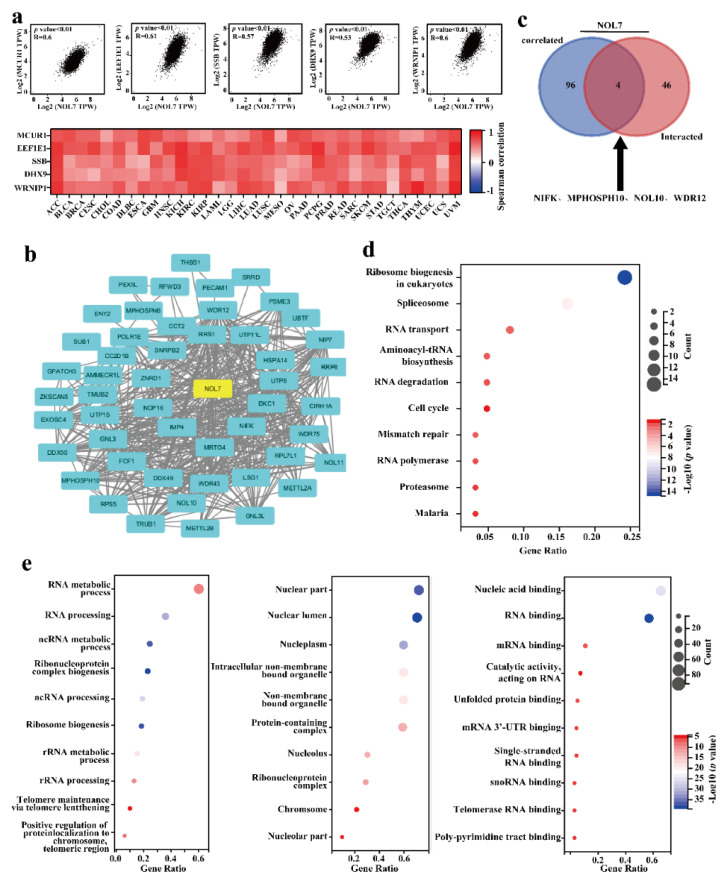
Enrichment analysis of *NOL7*-related genes. (**a**) The top 100 genes related to *NOL7* expression were obtained from the TCGA project, and the correlations between *NOL7* expression and selected target genes, including *MCUR1*, *EEF1E1*, *SSB*, *DHX9*, *WRNIP1*, and *ABT1*, were analyzed using GEPIA2. (**b**) A PPI network of 50 experimentally verified *NOL7*-interacting proteins. The corresponding heatmap data for the detailed cancer types are shown below. (**c**) An intersection analysis of the NOL7-interacting and NOL7-correlated genes was conducted, and four genes were obtained, including *NIFK*, *MPHOSPH10, NOL10*, and *WDR12*. (**d**) KEGG pathway analysis of *NOL7*-interacting genes and *NOL7*-correlated genes. (**e**) GO analysis of *NOL7*-interacting genes and *NOL7*-correlated genes. Enrichment results for biological processes (**left**), cellular component (**middle**), and molecular functions (**right**) are displayed.

## Data Availability

Please contact the author regarding data requests.

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
