# Peer review of "Pancancer Analysis of the Oncogenic and Prognostic Role of NOL7: A Potential Target for Carcinogenesis and Survival"

_ijms, 2022, doi:10.3390/ijms23179611_

Round 1
Reviewer 1 Report
NOL7 was reported as a candidate tumor-suppressor gene for cervical cancer in 2006, and the mechanism of carcinogenesis inhibition has since been elucidated through functional analysis of the NOL7 protein. On the other hand, the authors reported in 2021 that NOL7 acts as an oncogene that promotes carcinogenesis and metastasis in melanoma. Therefore, in this manuscript, the authors performed a pan-cancer analysis of the NOL7 gene by accessing databases such as TCGA and analyzing the expression of the NOL7 gene in available cancer tissues and their controls, and concluded that the NOL7 gene is a predictive target for carcinogenesis and cancer progression.
 The authors' attempt to show that NOL7 is an oncogenic gene by a pan-cancer analysis using a comprehensive cancer database is commendable. However, the conclusion that NOL7 is pro-carcinogenic in many carcinomas but inhibitory in some others shows that the scientific approach is superficial, and readers of this paper will not be convinced.
Below I list the individual points that I noticed. Please consider them when writing new papers in the future.
- - Line 45: NOL7 should be written in italics. (Gene names are the same below).
- - Line 68 (Supplementary Figure 1a): The location of the gene is not clearly indicated.
- - Lines 69-71 (Supplementary Figure 1bc): All the data are in the HPA database, and the phrase "we investigated the subcellular localization..." is inappropriate.
- - Lines 74-79 (Figure 1a): Although described as a comparison of tumor tissues and adjacent normal tissues, the tumors shown in Figure 1a all have different numbers of samples of tumor and normal tissues. This indicates that this is not a comparison of tumor tissues and adjacent normal tissues within the same individual. For analysis of genes such as NOL7, for which individual differences in expression levels are large, comparisons should be made between cancerous and surrounding noncancerous tissues within the same individual.
- - Line 85: Isn't the CHOL data shown in Figure 1b exactly the same as the data in Figure 1a?
- - Line 92: In Figure 1c, the expression of NOL7 is indeed decreased in PAAD, but in Figure 1b, it is increased in PAAD.
- - Lines 112-113: In KICH, KIRC, KIRP and THCA, NOL7 expression appears to decrease with cancer progression.
- - Line 219: CD274 may be CD276?
- - Lines 239-241: These are all things that have already been reported as the biological significance of NOL7. significance of NOL7.
- - Line 278: NOL8 may be NOL10?
Round 2
Reviewer 1 Report
I have no more comment.